# Inclusion of Medium-Chain Triglyceride in Lipid-Based Formulation of Cannabidiol Facilitates Micellar Solubilization In Vitro, but In Vivo Performance Remains Superior with Pure Sesame Oil Vehicle

**DOI:** 10.3390/pharmaceutics13091349

**Published:** 2021-08-27

**Authors:** Wanshan Feng, Chaolong Qin, Elena Cipolla, Jong Bong Lee, Atheer Zgair, Yenju Chu, Catherine A. Ortori, Michael J. Stocks, Cris S. Constantinescu, David A. Barrett, Peter M. Fischer, Pavel Gershkovich

**Affiliations:** 1School of Pharmacy, University of Nottingham, Nottingham NG7 2RD, UK; wanshan.feng@nottingham.ac.uk (W.F.); chaolong.qin1@nottingham.ac.uk (C.Q.); elena.cipolla94@gmail.com (E.C.); myjblee@gmail.com (J.B.L.); ph.atheer.zgair@uoanbar.edu.iq (A.Z.); yenju.chu@nottingham.ac.uk (Y.C.); cath.ortori@nottingham.ac.uk (C.A.O.); michael.stocks@nottingham.ac.uk (M.J.S.); david.barrett@nottingham.ac.uk (D.A.B.); peter.fischer@nottingham.ac.uk (P.M.F.); 2School of Pharmacy, Universita di Roma Tor Vergata, 00173 Rome, Italy; 3College of Pharmacy, University of Anbar, Ramadi 31001, Iraq; 4Tri-Service General Hospital, Medical Supplies and Maintenance Office, National Defense Medical Center, Taipei 114202, Taiwan; 5Division of Clinical Neuroscience, University of Nottingham, Nottingham NG7 2UH, UK; cris.constantinescu@nottingham.ac.uk

**Keywords:** cannabidiol, lipid-based formulation, lymphatic transport, sesame oil, surfactant, medium-chain triglyceride

## Abstract

Oral sesame oil-based formulation facilitates the delivery of poorly water-soluble drug cannabidiol (CBD) to the lymphatic system and blood circulation. However, this natural oil-based formulation also leads to considerable variability in absorption of CBD. In this work, the performance of lipid-based formulations with the addition of medium-chain triglyceride (MCT) or surfactants to the sesame oil vehicle has been tested in vitro and in vivo using CBD as a model drug. The in vitro lipolysis has shown that addition of the MCT leads to a higher distribution of CBD into the micellar phase. Further addition of surfactants to MCT-containing formulations did not improve distribution of the drug into the micellar phase. In vivo, formulations containing MCT led to lower or similar concentrations of CBD in serum, lymph and MLNs, but with reduced variability. MCT improves the emulsification and micellar solubilization of CBD, but surfactants did not facilitate further the rate and extent of lipolysis. Even though addition of MCT reduces the variability, the in vivo performance for the extent of both lymphatic transport and systemic bioavailability remains superior with a pure natural oil vehicle.

## 1. Introduction

Approximately 30% to 60% of new active drugs have poor aqueous solubility issues, and conventional formulations usually do not facilitate the absorption of these compounds from the gastrointestinal (GI) tract [1]. Lipid-based formulations have been proposed to facilitate the intraluminal solubility and systemic bioavailability of these poorly water-soluble compounds [1,2,3,4]. One of the most common lipidic core excipients used in various lipid-based formulations is different vegetable fats, for example sesame oil [5,6,7].

The main component in most natural vegetable oils is long-chain triglycerides (LCTs), which are digested in the intestinal lumen by lipases to generate free fatty acids and monoglycerides. Since the lipases act at the surface of the oil droplets, the emulsification step of the lipidic formulation in the intestinal tract is extremely important for the digestion of lipids and the absorption of the co-administered lipophilic drug. The triglyceride lipolysis products together with bile salts and phospholipids then form mixed micelles in the intestinal lumen. Mixed micelles facilitate the transport of the lipophilic compounds further across the unstirred aqueous layer and into the enterocytes. Triglycerides are then re-synthesized intracellularly from long-chain fatty acids and monoglycerides, and large lipoproteins (chylomicrons) are then assembled in the enterocytes. The chylomicrons are taken up by lymph lacteals rather than blood capillaries due to their large size [8,9]. The association of lipophilic compounds with chylomicrons in the enterocytes facilitates the intestinal lymphatic transport of drugs and avoids hepatic first pass metabolic loss.

Cannabidiol (CBD), a BCS class II drug, is a highly lipophilic (clogD_7.4_ 6.53) non-psychoactive phytocannabinoid, which has substantial first-pass metabolism contributing to low oral bioavailability [10,11]. For highly lipophilic drugs with logD_7.4_ above five, the intestinal lymphatic transport is frequently a useful drug delivery route to avoid hepatic first-pass metabolic loss and increase bioavailability, for example halofantrine, vitamin D_3_ and tetrahydrocannabinol (THC) [3,6,7,12,13,14].

CBD has therapeutic potential in the treatment of auto-immune and inflammatory diseases, such as multiple sclerosis (MS), rheumatoid arthritis (RA) and allergic asthma [15,16,17,18]. The immunosuppressive effects of CBD have been extensively studied, and it was found that CBD could suppress tumor necrosis factor (TNF) and interleukin (IL) cytokine production in both rats and human mononuclear cells [6,19,20,21]. It has been also shown that sesame oil vehicle facilitates the transport of CBD through the intestinal lymphatic system. When CBD was administered orally in natural sesame oil vehicle to rats, the systemic bioavailability of CBD has increased 2.8-fold compared to the lipid-free vehicle, and the drug concentration in lymph fluid was 250-fold higher compared to plasma [6,7]. A recent clinical study has also shown that sesame oil has increased CBD oral absorption eight-fold compared to CBD powder in healthy male volunteers [22]. However, both the intestinal lymphatic transport and systemic bioavailability of CBD administered in sesame oil vehicle were associated with substantial variability [5,6,7].

In the present study, we hypothesize that enhancement of the emulsification and micellar solubilization of CBD in the GI tract could contribute to reduction in the variability of the absorption of the drug. The rationale for this hypothesis comes from a previously conducted in vitro lipolysis study suggesting that less than 15% of triglyceride-C18 (Tri-C18) was digested in simulated human intestinal fluids [23]. On the other hand, the same lipolysis experiment showed that more than 90% of medium-chain triglyceride (MCT), such as triglyceride-C8 (Tri-C8), was hydrolyzed under the same conditions [23,24]. Thus, the addition of MCT to the sesame oil can potentially improve the CBD micellar solubilization following lipid digestion in the GI tract, and consequently enhance the drug absorption and the intestinal lymphatic transport.

To enhance further the emulsification and micellar solubilization, nonionic surfactants can be used. In this study, selection of surfactants was based on their hydrophilic–lipophilic balance (HLB) value. Surfactants with an HLB value ranging from 8 to 18 are suitable for the oil in water emulsification. Therefore, Tween 85, Tween 80 and Span 20 have been selected for this study [25,26]. In addition, a derivative of Vitamin E, d-α-tocopherol polyethylene glycol 1000 succinate (TPGS), which has been previously shown as a powerful solubilizer for lipophilic compounds, has also been selected [27,28,29]. The free α-tocopherol released from TPGS could also prevent the fatty acid oxidation process [30,31,32]. It has also been reported that TPGS can enhance lymphatic transport of lipophilic compounds by stimulating the chylomicron secretion in Caco-2 cells [33].

Therefore, the aim of this study is to test the hypothesis that the addition of MCT and surfactants to long-chain triglyceride vehicle (sesame oil) improves the emulsification and micellar solubilization of CBD in vitro, and eventually leads to increased intestinal lymphatic transport and drug bioavailability with reduced variability in vivo.

## 2. Materials and Methods

### 2.1. Materials

Sesame oil, glyceryl trioctanoate, polyoxyethylenesorbitan trioleate (Tween^®^ 85, Sigma-Aldrich, Dorset, UK), polyethylene glycol sorbitan monooleate (Tween^®^ 80), sorbitan monolaurate (Span^®^ 20, Sigma-Aldrich, Dorset, UK), d-α-Tocopherol polyethylene glycol 1000 (TPGS), sodium hydroxide solution (NaOH, 1 M), l-α-phosphatidylcholine (~60%, from egg yolk), Trizma^®^ (Sigma-Aldrich, Dorset, UK) maleate, sodium taurocholate hydrate, pancreatin from porcine pancreas (8 × USP specifications), sodium taurocholate hydrate, probucol, 4,4-dichlorodiphenyltrichloroethane (DDT) and serum triglyceride determination kit were all purchased from Sigma-Aldrich (Dorset, UK). Sodium chloride and anhydrous calcium chloride were purchased from Fisher Scientific (Leicester, UK). Cannabidiol (CBD, ≥98%) was purchased from THC Pharm (Frankfurt, Germany). Rat plasma was purchased from Sera Laboratories International (West Sussex, UK). All solvents were purchased from Fisher Scientific (Leicester, UK), and were of HPLC grade.

### 2.2. Lipid-Based Formulations

All lipid vehicles were prepared in the volumetric flask. Sesame oil and Tri-C8 mixture were firstly prepared, then surfactants were blended in a lipid mixture using the magnetic stirrer (1000 rpm) under 37 °C in a water bath until TPGS was fully dissolved. Lipid-based formulations were prepared by solubilizing CBD in the pre-mixed lipid vehicle under the same conditions as described above, then transferred into glass scintillation vials. Sesame oil was used as a vehicle for the control group, and as the LCT component in other formulations. In addition to sesame oil control group, five lipid-based formulations were prepared: Formulation 1 (F1, sesame oil:Tri-C8:Tween^®^ 80, 5:3:2, *v*/*v*/*v* with 10 mg/mL TPGS), Formulation 2 (F2, sesame oil:Tri-C8:Tween^®^ 85, 5:3:2, *v*/*v*/*v* with 10 mg/mL TPGS), Formulation 3 (F3, sesame oil:Tri-C8, 1:1, *v*/*v*), Formulation 4 (F4, sesame oil:Tri-C8:Tween^®^ 85, 2:2:1, *v*/*v*/*v* with 10 mg/mL TPGS) and Formulation 5 (F5, sesame oil:Tri-C8:Tween^®^ 85:Span^®^ 20, 5:3:1:1, *v*/*v*/*v*/*v* with 10 mg/mL TPGS) (Table 1). The concentration of CBD was 50 mg/mL in all formulations, which were freshly prepared on the day of the experiment for all in vitro and in vivo tests.

### 2.3. In Vitro Lipolysis

The lipid digestion process in the GI tract was assessed using the in vitro lipolysis system. The composition of simulated human intestinal digestion buffer has been reported before [23,34,35]. Briefly, the complete buffer consisted of 50 mM Trizma^®^ maleate, 150 mM NaCl, 5 mM CaCl_2_, 5 mM sodium taurocholate hydrate and 1.25 mM lecithin in water to mimic fasted state intestinal fluids. The pancreatic lipase extract was prepared in the incomplete digestion buffer, which was composed of 50 mM Trizma^®^ maleate, 5 mL 150 mM NaCl and 5 mM CaCl_2_. One gram pancreatin was blended into incomplete digestion buffer and vortexed for 15 min at room temperature. The supernatant of lipase extract was collected after centrifugation at 5 °C, 3000 rpm for 15 min, and then stored on ice.

Per oral medicines are assumed to be taken together with approximately 250 mL of water. The total volume used for in vitro lipolysis was scaled down 10 times compared with human parameters. Briefly, 22.2 mL complete digestion buffer containing 0.3 mL lipid formulation was pre-mixed at 37 °C in a water bath for 15 min. Pancreatic lipase extract (2.2 mL) was then added to the complete buffer to start the lipolysis reaction. The digestion process was monitored by the pH titration system (LabX light v3.1, Mettler-Toledo Limited, Leicester, UK), and the reaction was assumed as complete when addition rate of 1 M NaOH was slower than 3 μL/min.

The final lipolysis medium was separated into three layers (sediment, micellar and undigested lipid layer) by means of ultra-centrifugation at 268,350× *g*, 37 °C for 90 min [34,36,37,38]. All three fractions have been stored at −80 °C until analysis.

### 2.4. Animal Studies

Animal welfare and all experimental procedures were reviewed and approved by the University of Nottingham Ethical Review Committee under the Animals (Scientific Procedures) Act 1986. Male Sprague Dawley rats (340–380 g, Charles River Laboratories, Margate, UK) were used for in vivo studies. All animals were housed in the University of Nottingham Bio Support Unit under the conditions of controlled temperature and humidity. The rats were provided with free access to water and food with a 12-h light–dark cycle. As a preparation for the pharmacokinetic study, rats underwent jugular vein cannulation surgery and were allowed to recover for two nights before the pharmacokinetics experiment. Animals were fasted overnight up to 16 h before the pharmacokinetic experiment.

Three formulations were administered by oral gavage to animals, including sesame oil formulation, Formulation 3 (Sesame oil:Tri-C8, 1:1, *v*/*v*) and Formulation 4 (Sesame oil:Tri-C8:Tween^®^ 85, 2:2:1, *v*/*v*/*v* and 10 mg/mL TPGS). All formulations contained 50 mg/mL CBD and were administered at a dose of 25 mg/kg.

For the pharmacokinetic study, 0.2 mL blood was withdrawn from jugular vein cannula at 0.5, 1, 1.5, 2, 2.5, 3, 4, 6, 8 and 12 hours’ time following drug administration and collected into K3 EDTA-containing Eppendorf tubes. After centrifugation (3000× *g* for 10 min), plasma was stored at –80 °C until analysis.

Biodistribution studies were carried out based on the plasma t_max_ determined in pharmacokinetic studies. All rats were housed with free access to water and fasted overnight up to 16 h before the biodistribution experiment. All formulations were prepared on the day of experiment and administered in the same manner as for the pharmacokinetic studies. All rats were sacrificed at 1.5 or 2.5 h post-administration in all groups. Blood, lymph fluid and mesenteric lymph nodes (MLN) were then harvested as described in our previous studies and stored at –80 °C until analysis [5,6]. In addition, after the completion of pharmacokinetic study at 12 h, animals were sacrificed, and MLN were collected for further analysis.

### 2.5. Bioanalytical Conditions

#### 2.5.1. Sample Preparation

The sample preparation procedure was performed using a combination of protein precipitation and liquid–liquid extraction before injection into the HPLC system as previously described [5]. Briefly, fifty microliters of lipid phase, 100 μL of micelle phase and 100 μL sediment phase were used for sample preparation in a 16 × 100 mm glass tube. Ten microliters of stock solution of probucol (IS) in acetonitrile at a concentration of 2 mg/mL was spiked into lipid phase, and 1 mg/mL into micelle and sediment phases. Then, 0.3 mL tetrahydrofuran and 3 mL *n*-hexane were added into the test tube to extract CBD. After 3 min of vortex, the test tubes were centrifuged at 1160× *g* for 10 min at room temperature. The upper organic layer was transferred to a clean test tube and evaporated under a gentle stream of nitrogen gas (Techne Dri-Block Sample Concentrator, Cambridge, UK) at 35 °C to dryness. The dry residual was then reconstituted in 1 mL acetonitrile for all phases, and 20 μL of the sample was injected into the HPLC system.

The sample preparation method for plasma (or serum) and tissue samples was performed as previously described [5,39]. Briefly, MLNs were isolated from the surrounding fatty tissues, then homogenized in water in ratio of 1:3 (*w*/*v*). The lymph fluid sample was diluted 10 times in blank rat plasma. The volume used for sample preparation of rat plasma, homogenized MLNs and diluted lymph fluid was 100 μL. Ten microliters of DDT stock solution in acetonitrile (IS, 50 μg/mL) was spiked into samples. The cold acetonitrile (450 μL) was added for protein precipitation and the samples were vortexed for 5 min. Water (450 μL) and 3 mL *n*-hexane were then added, followed by vortex-mixing for 5 min. The rest of procedures were the same as for lipolysis fractions samples, but the dry residue was reconstituted in 100 μL ACN.

#### 2.5.2. Chromatography Conditions

The analysis of CBD in in vitro lipolysis fractions, rat plasma, rat lymph and lymph nodes was performed by means of a validated high-performance liquid chromatography (HPLC) method [5,39]. The system consisted of Waters 600 Pump, Waters 717 Autosampler and Waters 2996 Photodiode Array Detector. The CBD and internal standard (IS) were detected at 220 nm wavelength.

For lipolysis fractions samples, the separation was achieved using an ACE Excel Super C18 100 × 4.6 mm, 5 μm particle size column, protected by an ACE C18-PFP 3 μm guard cartridge, as previously reported. The mobile phase was a mixture of acetonitrile and water in a ratio of 92:8 (*v*/*v*) at isocratic conditions. The flow rate was set at 0.6 mL/min and 43 °C for column temperature [39].

For rat plasma, serum and tissue separation was achieved using an ACE C18-PFP 150 × 4.6 mm, 3 μm column, protected by an ACE C18-PFP 3 μm guard cartridge, as previously reported. The mobile phase consisted of acetonitrile: water (62:38, *v*/*v*) at isocratic conditions. Flow rate was set at 1 mL/min, column temperature was maintained at 55 °C, and injection volume was 40 μL [5,39].

### 2.6. Triglycerides Level Determinations

The triglyceride level measurement was performed using a commercially available kit (TR0100, Sigma, Gillingham, UK) following the manufacturer’s instructions.

### 2.7. Data Analysis

Phoenix WinNonlin 6.3 Professional (Pharsight, Mountain View, CA, USA) was applied for non-compartmental pharmacokinetic analysis of the obtained plasma concentrations. All data were presented as mean ± standard deviation (SD). One-way analysis of variance (ANOVA), followed by Dunnett’s post-hoc comparison was used for statistical analysis. P values of less than 0.05 were considered statistically significantly different. Statistical analysis was performed using GraphPad Prism version 7.0d (GraphPad software, San Diego, CA, USA).

## 3. Results

### 3.1. In Vitro Lipolysis of Lipid-Based Formulations

The processing of lipid-based formulations in the intestinal lumen has been assessed using in vitro lipolysis system (Figure 1). The CBD distributed into the micelle phase represents the fraction readily available for absorption from the small intestine [34]. Figure 1 clearly shows that addition of the MCT resulted in a trend for higher CBD distribution into the micellar phase. Moreover, when the ratio of Tri-C8 and sesame oil in Formulation 3 and Formulation 4 reached 1:1, the micellar phase had a statistically significantly higher amount of CBD than the sesame oil group.

The recovery of triglyceride, diglyceride and monoglyceride in lipolysis phases has been calculated and shown in Figure 2. The lipid fractions in the micellar phase are mainly diglycerides or monoglycerides, and significantly higher levels of these lipids have been found in the sesame oil group compared to other five formulations in this study. The lipid phase represents the proportion of lipidic formulation that has not been hydrolyzed completely, and the sesame oil group has a significantly higher amount of triglycerides in the lipid phase compared to Formulations 3, 4 and 5.

### 3.2. In Vivo Pharmacokinetics

Formulations 3 and 4 were selected to proceed to in vivo pharmacokinetic and biodistribution studies due to higher proportion of the drug distributed to the micellar phase in in vitro lipolysis experiments. Pure sesame oil vehicle was also selected for in vivo studies to serve as a control group. The plasma concentration–time profiles of CBD following oral administration of these formulations are presented in Figure 3. The pharmacokinetic parameters including half-life, t_max_, C_max_ and the plasma concentration-time curve (AUC) are outlined in Table 2. The AUC of CBD concentrations for sesame oil formulation are significantly higher than the Formulation 3. However, there are no statistically significant differences for other parameters when Formulations 3 and 4 are compared to the sesame oil group.

### 3.3. Biodistribution

The blood, lymph fluid and MLNs were collected in the biodistribution experiments at pre-determined time points. The time points chosen are based on the in vivo plasma pharmacokinetics results and cover the early and late periods in the absorption phase of CBD. The triglyceride levels have also been measured in serum and lymph fluid samples, and are presented together with CBD concentrations in Figure 4 and Figure 5. There are no statistically significant differences in triglyceride levels at both the 1.5 and 2.5 h time points, when Formulations 3 and 4 are compared to the sesame oil group (for both serum (Figure 4) and lymph (Figure 5) samples). It has been found that CBD concentration in serum at 2.5 h following oral administration of pure sesame oil-based formulation is significantly higher than after administration of Formulation 4 (Figure 4b). For CBD concentrations in the lymph fluid, there are no significant differences between Formulations 3 or 4 when they are compared to sesame oil at 1.5 and 2.5 h (Figure 5). The administration of CBD in sesame oil leads to a higher concentration of CBD in MLNs at 2.5 h compared to Formulation 4 (Figure 6b), whereas no differences were found at 1.5 h (Figure 6a). The MLNs have been also collected at the 12 h time point at the end of the in vivo pharmacokinetic study, and the levels of CBD following administration of sesame oil-based formulation were significantly higher than for Formulation 4 (Figure 6c).

## 4. Discussion

Even though sesame oil-based formulation facilitated the oral bioavailability and intestinal lymphatic transport of CBD in previous studies, the substantial inter-subject variability is a major issue which needs to be addressed [6,7]. The pre-digested lipid-based formulations have been previously suggested to minimize such variability by completely avoiding the LCT digestion step in the intestinal lumen. However, it was shown in our previous work that the pre-digested lipids did not reduce the variability or enhance the intestinal lymphatic transport or systemic bioavailability of CBD in comparison to the sesame oil vehicle [5].

Multiple previous works have suggested that the addition of the MCTs into lipid-based formulations can enhance the emulsification and micellar solubilization, and eventually the bioavailability of lipophilic drugs [3,40,41,42,43,44]. Previously reported in vitro lipolysis data also suggest that the MCTs are digested more readily in the simulated human intestinal fluids compared to the LCTs [23,24,35,45]. Moreover, addition of surfactants into a lipid-based formulation can enhance the micellar emulsification in the GI tract. Therefore, addition of the MCT and surfactants to the natural sesame oil has been attempted in this study for improvement of micellar solubilization of CBD and reduction of the in vivo variability in bioavailability and lymphatic transport observed with sesame oil vehicle.

### 4.1. The Design of Lipid-Based Formulations

When lipid-based formulations are designed for a highly lipophilic compound such as CBD, with proven substantial intestinal lymphatic transport, a certain proportion of LCT must remain in the formulation to maintain the intestinal lymphatic transport element of the absorption [5,6,7]. Therefore, the proportion of sesame oil was maintained at 40% to 50% of the total lipid volume in Formulations 1–5. The dose and concentration of CBD in the formulations were dominated by two major factors: (1) the clinical oral therapeutic dose range of CBD in humans, and (2) the size range of oral soft gelatin capsules suitable for oral administration.

CBD and its therapeutic potential have been studied for decades. CBD has been suggested as a potential therapeutic agent for diverse medical conditions, including autoimmune disease, inflammations, cancer or schizophrenia. The reported dose range in humans for different conditions is also very wide, from 16 mg to 3000 mg per day [46,47,48,49]. CBD has an inverted U-shaped dose–response curve in animal models and human volunteers, with the most effective single oral dose in humans being around 300 mg [49,50,51].

The largest soft gelatin capsules that can be found on the market are 25 oblong capsules, which contain approximately 1.5 mL (for example 1500 mg Evening primrose oil and Starflower oil produced by Holland and Barrett (Hinckley, UK)). Therefore, the concentration of CBD in formulations in this study (50 mg/mL) is based on mimicking realistic human use conditions of 300 mg CBD dose in a 3 mL (two 25 oblong capsules) lipid vehicle.

To note, the original concentration of CBD considered was 100 mg/mL (to mimic one 25 oblong capsule administration). However, our previous work showed that CBD concentration above 80 mg/mL in lipid-based formulations could affect the digestion process during in vitro lipolysis [52].

The rat model was utilized to assess the lipid-based formulations in this study. The dose of CBD was calculated in rats using allometric scaling from the human dose. Assuming a 300 mg CBD dose in a 70 kg adult human, the allometrically scaled dose in rats is 26 mg/kg [53]. Therefore, a dose of 25 mg/kg CBD was administered to rats in this study.

### 4.2. In Vitro Lipolysis

Based on the in vitro lipolysis results in this study, mixing MCT with sesame oil, as predicted, indeed leads to a trend of higher CBD distribution into the aqueous micellar phase compared to the sesame oil vehicle (Figure 1). The statistical significance has been found for Formulations 3 and 4, when compared to the sesame oil-based formulation (control). Unlike Formulations 1, 2 or the sesame oil vehicle, the proportion of Tri-C8 is 50% of total lipids for Formulation 3 and Formulation 4 (Table 1). Such a high ratio of Tri-C8 in these formulations results in overall more efficient digestion during the in vitro lipolysis process.

Lower levels of triglycerides remain in the lipid phase for Formulations 3, 4, and 5 compared to the sesame oil group following in vitro lipolysis (Figure 2). This suggests that the higher proportion of Tri-C8 indeed leads to more efficient lipid digestion and a higher CBD distribution into the micellar phase for Formulations 3 and 4 compared to the sesame oil (Figure 1 and Figure 2). In addition, the measured “triglyceride” (free glycerol and glycerol released from triglyceride, diglyceride and monoglyceride) levels in the micellar phase also suggest that addition of Tric-C8 to the formulation enhances the lipid digestion and micellar solubilization of CBD in comparison to sesame oil vehicle.

The lipolysis results indicate that addition of surfactants (HLB value from 8.6 to 15) affects lipid digestion in vitro. During the digestion of lipids, the surfactants in lipid-based formulations are readily displaced from the lipid droplet surface by bile acids [54,55]. This allows lipase to approach the interface of the oil-in-water emulsion and initiate lipolysis [56]. The hydrolyzed fatty acids and 2-monoglycerides compete with the surfactants at the lipid droplet surface. The excess of fatty acids and 2-monoglycerides subsequently form mixed micelles with bile salts and surfactants. However, there are in vitro and in vivo studies indicating that highly hydrophilic surfactants (HLB 12-17) can inhibit pancreatic lipase activity and therefore reduce the hydrolysis of triglycerides, resulting in lower oral bioavailability of the poorly water-soluble drugs [57,58,59]. In agreement between these studies and the current study, Formulation 3 contains no surfactants but has shown the most effective lipid digestion compared to other formulations based on the *in vitro* lipolysis results (Table 1, Figure 1 and Figure 2). Therefore, the addition of surfactants into sesame oil did not improve triglyceride hydrolysis in the in vitro lipolysis system in this study.

### 4.3. In Vivo Pharmacokinetics and Biodistribution

Based on the in vitro lipolysis results, sesame oil and Formulations 3 and 4 were selected to proceed further for in vivo pharmacokinetics study. However, unlike the in vitro lipolysis, the in vivo pharmacokinetic results have shown that the systemic bioavailability of CBD following oral administration of sesame oil formulation is higher than for Formulation 3, and similar to the Formulation 4 (Table 2).

There is, therefore, a lack of correlation between in vitro and in vivo results in this study. Potential explanation could be related to the fact that there are two steps that dominate the bioavailability of CBD, the lipid digestion in the intestinal lumen leading to intraluminal micellar solubilization of CBD, and the intestinal lymphatic transport (which is related to the association of CBD with chylomicrons in the enterocyte). The in vitro lipolysis model simulates only one step out of these two—the lipid digestion in the intestinal lumen. A similar lack of in vitro/in vivo correlation has been previously reported for other drugs with a substantial intestinal lymphatic transport component in the absorption process [24,60]. The lipid digestion and chylomicron formation in the enterocyte are both complex processes, and there is currently no appropriate single in vitro model that can simulate the entire process, including intraluminal and intracellular events. It has been suggested that a better prediction of bioavailability of highly lipophilic drugs administered in lipid-based formulations could be achieved by combining the in vitro lipolysis with a microsome metabolism assay [61,62]. It is possible that in the future, affinity to chylomicrons should be added to in vitro lipolysis and microsomal stability assays for better prediction of bioavailability of highly lipophilic drugs administered in lipidic formulations [12,63].

The area under the curve (AUC) values indicate that Formulations 3 and 4 lead to lower variability in systemic exposure to CBD compared to the sesame oil vehicle (Table 2). A high proportion of Tri-C8 facilitated the digestion of lipids and enhanced the micellar solubilization of CBD in the intestinal lumen. Formulation 3 has a slightly faster t_max_ than the sesame oil formulation, which could be due to faster digestion of Tri-C8 compared to sesame oil. However, chylomicrons mainly consist of LCTs, whereas MCTs are mostly transported through the portal vein rather than packed into chylomicrons in the enterocytes [3,24,45]. Therefore, a rapid digestion of Tris-C8 could deliver a part of CBD into enterocyte at the earlier digestion stage, before long-chain lipids become available in the enterocyte for formation of chylomicrons. Consequently, this quickly absorbed part of the CBD dose is likely to be delivered to the systemic circulation through portal vein with substantial hepatic first-pass metabolism, therefore resulting in reduced overall systemic exposure.

Indeed, addition of Tris-C8 into sesame oil reduced the AUC variability compared to pure sesame oil formulation. However, the extent of CBD absorption in vivo was not improved. As stated above, the surfactants with HLB in a range of 12 to 17 may negatively affect the lipolysis process, and therefore, decrease the drug distribution into the mixed micelles in the intestinal lumen. The TPGS is a surfactant with antioxidant properties, which has been reported to enhance chylomicron formation, as well as potentially being delivered into the intestinal lymphatic system by association with the chylomicrons [33]. However, it has also been reported that TPGS and oleic acid-containing mixed micelles have a negative effect on the bioavailability and lymphatic transport of the antiretroviral drug saquinavir in comparison to oleic acid microemulsions [64]. Another work suggested that TPGS was not hydrolyzed by pancreatic lipase in the GI tract and remains in the intact form until it transports to the enterocyte [58]. Therefore, it is possible that addition of the TPGS restricted the drug association with chylomicrons in the current study.

There are no significant differences in the triglyceride levels in both serum and lymph fluid samples, when Formulations 3 and 4 are compared to sesame oil formulation (Figure 4 and Figure 5). The triglyceride levels in serum and lymph fluid samples indicate that there might be an overload of LCT in these formulations for lymphatic transport, therefore resulting in the saturation in the process of triglyceride re-synthesis in the enterocytes. Even though both Formulations 3 and 4 contain approximately 50% of sesame oil compared to pure sesame oil vehicle formulation, the LCT uptake for these two formulations is similar to the sesame oil.

The overall results of the biodistribution studies suggest that sesame oil has a more efficient performance in enhancing the intestinal lymphatic transport of CBD compared to both Formulations 3 and 4.

Taken together, the combined results of in vivo pharmacokinetics and biodistribution studies suggest that addition of surfactants and MCT to natural sesame oil does not improve CBD oral bioavailability and intestinal lymphatic transport compared to simple sesame oil-based formulation.

## 5. Conclusions

Sesame oil-based formulation has been previously reported to lead to substantial increase in bioavailability and intestinal lymphatic transport of CBD in rats, but with considerable variability. The MCT incorporated into sesame oil enhanced the micellar solubility of CBD in vitro, but surfactants used in this study may have reduced the lipolysis of triglycerides in comparison to other MCT-containing formulations. Even though the addition of MCT into lipid-based formulations reduces the viability, pure sesame oil vehicle was still superior in the extent of CBD lymphatic transport and bioavailability in vivo. There are multiple factors that dominate the lipid digestion, including the chain length of triglyceride, or lipid interaction with surfactants. Furthermore, the lymphatic transport requires a combination of a highly lipophilic compound (with affinity to chylomicrons) and the presence of lipids that eventually lead to chylomicron formation (LCT or long-chain fatty acids). Further studies will be needed to assess alternative approaches to reduce inter-subject variability associated with pure sesame oil formulation of CBD without negatively affecting the extent of absorption of the drug.

## Figures and Tables

**Figure 1 pharmaceutics-13-01349-f001:**
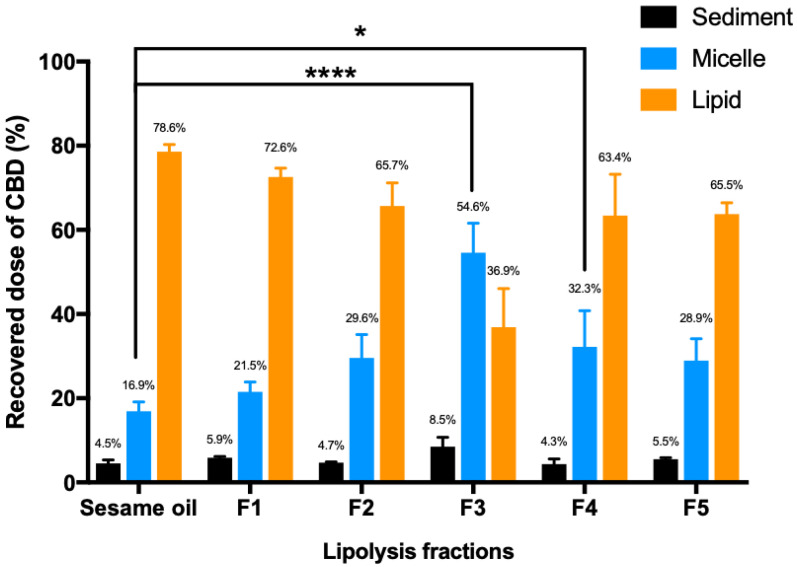
Distribution of CBD in sediment, micelle and lipid phases (*n* = 3, mean ± SD) following the in vitro lipolysis of 6 lipid-based formulations of CBD. One-way ANOVA was used for statistical analysis, followed by Dunnett’s post-hoc test with sesame oil serving as a control group. *, *p* < 0.05, ****, *p* < 0.0001. The experiment was terminated when the addition rate of NaOH was slower than 3 μL/min.

**Figure 2 pharmaceutics-13-01349-f002:**
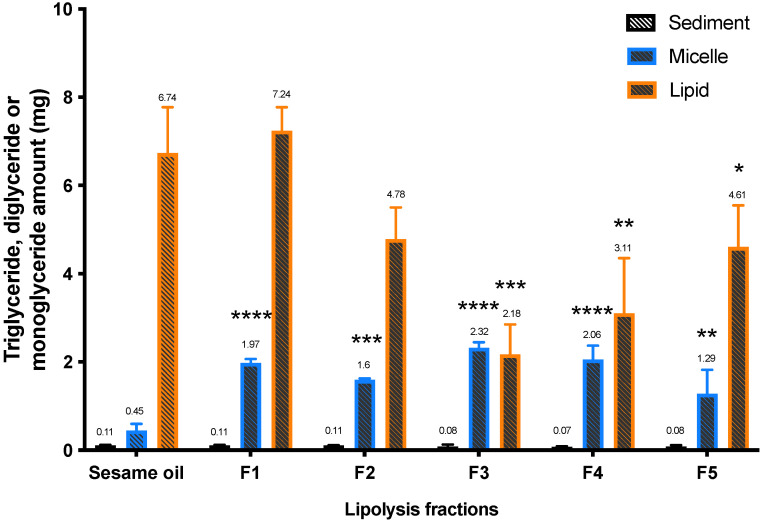
The amount of triglyceride, diglyceride and monoglyceride in lipolysis fractions following in vitro lipolysis of six lipid-based formulations (*n* = 3, mean ± SD). One-way ANOVA was used for statistical analysis, followed by Dunnett’s post-hoc test with sesame oil serving as a control group. * *p* < 0.05, **, *p* < 0.01, ***, *p* < 0.001, ****, *p* < 0.0001.

**Figure 3 pharmaceutics-13-01349-f003:**
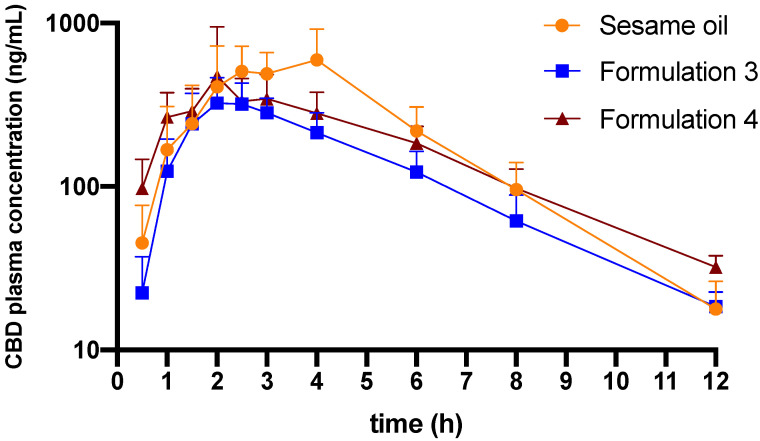
Plasma concentration-time profiles of CBD following oral administration of sesame oil-based formulation, Formulation 3 (sesame oil:Tri-C8, 5:5, *v*/*v*) and Formulation 4 (sesame oil:Tri-C8:Tween^®^ 85, 2:2:1, *v*/*v*/*v* with 10 mg/mL TPGS) (*n* = 5–6, mean ± SD). The concentration of CBD in all formulations was 50 mg/mL, the administered dose of CBD was 25 mg/kg in all treatment groups.

**Figure 4 pharmaceutics-13-01349-f004:**
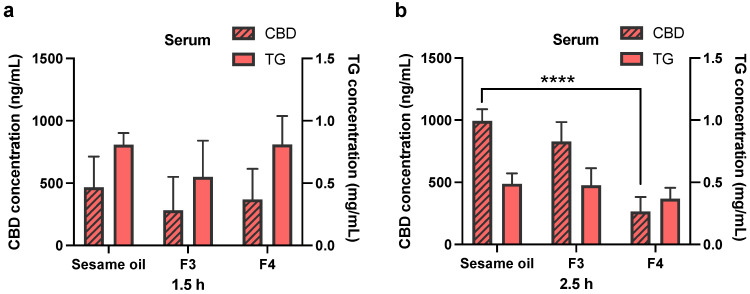
Concentrations of CBD and triglyceride in rat serum. CBD was orally administered in sesame oil, Formulation 3 (F3, sesame oil:Tri-C8, 5:5, *v*/*v*) and Formulation 4 (F4, sesame oil:Tri-C8:Tween 85, 4:4:2, *v*/*v*/*v* with 10 mg/mL TPGS) at a dose of 25 mg/kg in rats (50 mg/mL CBD content in each formulation). (**a**) The concentration of CBD and triglyceride level in rat serum at 1.5 h post-administration. (**b**) The concentration of CBD and triglyceride level in rat serum at 2.5 h post-administration. All data are shown as mean ± standard deviation (SD) (*n* = 4). Statistical analysis was performed using one-way ANOVA, followed by Dunnett’s post-hoc test with sesame oil serving as a control group. ****, *p* < 0.0001.

**Figure 5 pharmaceutics-13-01349-f005:**
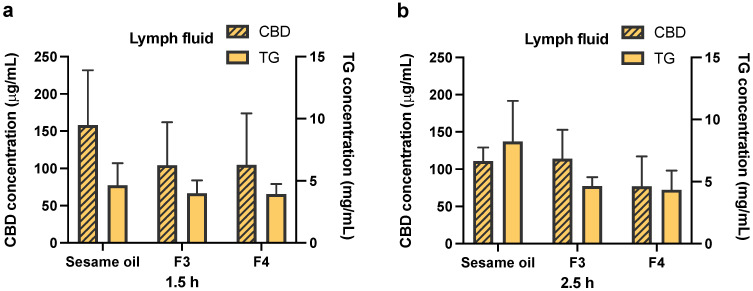
Concentrations of CBD and triglyceride in lymph fluid. CBD was orally administered in sesame oil, Formulation 3 (F3, sesame oil:Tri-C8, 5:5, *v*/*v*) and Formulation 4 (F4, sesame oil:Tri-C8:Tween 85, 4:4:2, *v*/*v*/*v* with 10 mg/mL TPGS) at a dose of 25 mg/kg in rats (50 mg/mL CBD content in each formulation). (**a**) The concentration of CBD and triglyceride in lymph fluid at 1.5 h post-administration. (**b**) The concentration of CBD and triglyceride in lymph fluid at 2.5 h post-administration. All data are shown as mean ± SD, *n* = 4. Statistical analysis was performed using one-way ANOVA followed by Dunnett’s post-hoc test with sesame oil serving as a control group.

**Figure 6 pharmaceutics-13-01349-f006:**
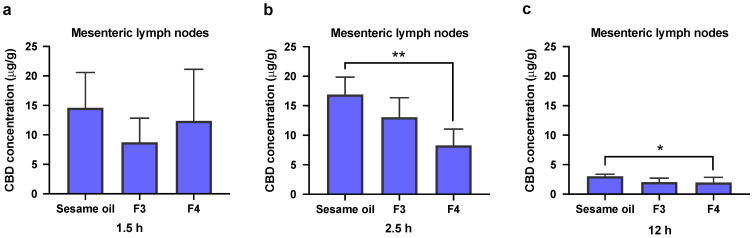
Concentrations of CBD in mesenteric lymph nodes (MLNs). CBD was orally administered in sesame oil, Formulation 3 (F3, sesame oil:Tri-C8, 5:5, *v*/*v*) and Formulation 4 (F4, sesame oil:Tri-C8:Tween 85, 4:4:2, *v*/*v*/*v* with 10 mg/mL TPGS) at a dose of 25 mg/kg in rats (50 mg/mL CBD content in each formulation). (**a**) The concentration of CBD in MLNs at 1.5 h post-administration. (**b**) The concentration of CBD level in MLNs at 2.5 h post-administration. (**c**) The concentration of CBD in MLNs at 12 h post-administration. All data are shown as mean ± SD, *n* = 4. Statistical analysis was performed using one-way ANOVA followed by Dunnett’s post-hoc test with sesame oil serving as a control group. *, *p* < 0.05, **, *p* < 0.01.

**Table 1 pharmaceutics-13-01349-t001:** The composition of the developed lipid-based formulations.

Formulation No.		Control	F1	F2	F3	F4	F5	HLB
Lipid	Sesame oil (*v*/*v*, %)	100	50	50	50	40	50	7
	Tri-C8 (*v*/*v*, %)	-	30	30	50	40	30	7
Surfactant	Tween^®^ 85 (*v*/*v*, %)	-	-	20	-	20	10	11
	Tween^®^ 80 (*v*/*v*, %)	-	20	-	-	-	-	15
	Span^®^ 20 (*v*/*v*, %)	-	-	-	-	-	10	8.6
	TPGS (mg/mL)	-	10	10	-	10	10	13.2

**Table 2 pharmaceutics-13-01349-t002:** Pharmacokinetic parameters of cannabidiol (CBD) derived from plasma concentration–time profiles following oral administration in different lipid-based formulations (mean ± SD (*n* = 5–6)).

Administration/ Formulation	t_1/2_ ^a^ (h)	t_max_ ^b^ (h)	C_max_ ^c^ (ng/mL)	AUC_0-∞_ ^d^ (h×ng/mL)	*n*
Sesame oil	1.6 ± 0.1	2.5–4	724 ± 318	2702 ± 909	5
Formulation 3	2.3 ± 0.3	2–2.5	371 ± 85	1512 ± 224 **	6
Formulation 4	2.5 ± 0.5	2–3	562 ± 390	2131 ± 373	6

^a^ Half-life; ^b^ time to maximum concentration in plasma; ^c^ The maximum concentration in plasma; ^d^ AUC from 0 to infinity; **, statistically significantly different from sesame oil (**, *p* < 0.01).

## Data Availability

All the data regarding the study is available through this manuscript.

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
