# Peer review of "Inclusion of Medium-Chain Triglyceride in Lipid-Based Formulation of Cannabidiol Facilitates Micellar Solubilization In Vitro, but In Vivo Performance Remains Superior with Pure Sesame Oil Vehicle"

_pharmaceutics, 2021, doi:10.3390/pharmaceutics13091349_

Round 1

Reviewer 1 Report

find this work highly important in the field of lipid based drug delivery systems. Therefore, I suggest to accept it for publication in the JOURNAL as is.

Author Response

Reviewer #1:

find this work highly important in the field of lipid based drug delivery systems. Therefore, I suggest to accept it for publication in the JOURNAL as is.

Response: We would like to thank the Reviewer for these kind comments.

Reviewer 2 Report

The paper submitted by Feng et al. deals with the preparation of a series of cannabidiol lipid-based formulations and their in vitro and in vivo characterization as compared to pure sesame oil solutions containing cannabidiol.

The manuscript is clear, well written and the conclusions are supported by the results. However, some corrections are needed in order to increase the overall quality of the paper:

  1. The title of section 2.2 is identical to that of section 2.3
  2. The section 2.2 must be completed. The authors must describe how the solubilisation of surfactants and cannabidiol were made (temperature, rpm, and duration). Moreover, the solubility of cannabidiol in sesame oil and MCT must be provided. Which is the final volume of the formulations?! Which is the concentration of surfactants in the final formulations?!
  3. The HLB values must be added for each surfactant in table 1
  4. Line 252: due to the higher distribution of the drug into the micellar phase…

Author Response

Reviewer #2:

The paper submitted by Feng et al. deals with the preparation of a series of cannabidiol lipid-based formulations and their in vitro and in vivo characterization as compared to pure sesame oil solutions containing cannabidiol.

The manuscript is clear, well written and the conclusions are supported by the results. However, some corrections are needed in order to increase the overall quality of the paper:

  • Reviewer comment: The title of section 2.2 is identical to that of section 2.3
  1.  

Response: As suggested by Reviewer, the title for section 2.3 has been changed to “In vitro lipolysis” in revised manuscript.

  • Reviewer comment: The section 2.2 must be completed. The authors must describe how the solubilisation of surfactants and cannabidiol were made (temperature, rpm, and duration). Moreover, the solubility of cannabidiol in sesame oil and MCT must be provided. Which is the final volume of the formulations?! Which is the concentration of surfactants in the final formulations?!

Response: As suggested by Reviewer, more details have been provided in the Revised Manuscript (lines 175-179). The CBD is a very lipophilic drug and is well soluble in MCT and LCT. The study by ‘D. Vaughn, J. Kulpa, L. Paulionis, Preliminary Investigation of the Safety of Escalating Cannabinoid Doses in Healthy Dogs, Front. Vet. Sci. 7 (2020) 1–13. https://doi.org/10.3389/fvets.2020.00051.’ was using 18.3 mg/mL CBD in MCT. Moreover, 160 mg/mL CBD solution in sesame oil has been used in ‘A. Zgair, Intestinal lymphatic transport of cannabinoids: implications for people with autoimmune diseases and immunocompromised individuals, University of Nottingham, 2017.’ In our current study, we used at least 40% sesame oil in the lipid-based formulation and the drug concentration was 50 mg/mL. Therefore, certainly CBD is well soluble in the lipid vehicles in this study. The concentration of surfactants in final formulation is shown in text lines 183-187 and also in Table 1 of the revised Manuscript.

  • Reviewer comment: The HLB values must be added for each surfactant in table 1

Response: As suggested by the Reviewer, the HLB values have been added in Table 1

  • Reviewer comment: Line 252: due to the higher distribution of the drug into the micellar phase…

Response: As suggested by the Reviewer, we have modified this sentence to ‘due to higher proportion of the drug distributed to the micellar phase’ in the revised manuscript (Line 341 of the revised Manuscript).

Reviewer 3 Report

The work describes the failure of the attempt to increase the oral bioavailability of CBD. This is the second tentative, as the Authors already tried with pre-digested or purified triglyceride based formulations, but not even that time they succeed in improving the lymphatic transport and bioavailability of CBD. The Authors performed very simple formulations of low novelty and interest, however the collected data could deserve to be published after some adjustments.

The reference list contains 13 citations before 2006, I think that more recent publications could be provided in most cases.

I suggest the Authors to perform nanoemulsions next time.

Specific comments:

  1. line 7, please check the author list
  2. line 39-41, the first three references are not recent (2003, 2007); the advancements in pharmaceutical technology in the field of drug delivery systems are in continuous development, therefore more up to date reviews should be included.
  3. line 42, the Authors state that sesame oil is the most common oil used as excipients for lipid-based formulations. However, the reported references deal only with CBD containing formulations, but they did not mentioned CBD in the text yet. I suggest the Author to describe more in depth what lipid formulations are, which oils are used, providing examples of their employment with different drugs.
  4. line 61, please provide examples of drugs whose bioavailabilities were increased by lymphatic transport, and add some references.
  5. line 73, please described the most popular CBD formulations and the latest advancements present in the literature, highlighting the drawbacks and the positives.
  6. line 90, please add more recent works regarding the solubilizing efficiency of vitamin E TPGS (i.e. retinoic acid, resveratrol).
  7. line 115, LCT is right or is MCT?
  8. Figure 2. lines 238-244. The discussion is not clear, the total amounts of tri-di-monoglycerides are different in each formulations as obviously their proportion is different in the related starting mixtures. The data should be provided considering their concentration in each formulation (recovered mg / initial mg x100).

Author Response

Reviewer #3:

  • Reviewer comment: The work describes the failure of the attempt to increase the oral bioavailability of CBD. This is the second tentative, as the Authors already tried with pre-digested or purified triglyceride based formulations, but not even that time they succeed in improving the lymphatic transport and bioavailability of CBD. The Authors performed very simple formulations of low novelty and interest, however the collected data could deserve to be published after some adjustments.

Response: We thank the reviewer for this comment. We are an academic research lab, and trying to understand the direction of the effect of different excipients on the lymphatic transport in bioavailability of a model highly lipophilic drug CBD, rather than developing a commercial formulation without knowing what factors contributed to its performance. It is more important for us to understand the nature of the effect of different excipients rather than to come up with a formulation recipe that performs better than sesame oil for reasons we cannot explain.

  • Reviewer comment: The reference list contains 13 citations before 2006, I think that more recent publications could be provided in most cases.

Response: We respectfully disagree with this comment of the Reviewer about older references. There are some very good papers published before 2006 and still relevant. We have therefore added more recent references without removing the older one. For example, line 41 the following more recent reference have been added: V. Patel, R. Lalani, D. Bardoliwala, S. Ghosh, A. Misra, Lipid-Based Oral Formulation Strategies for Lipophilic Drugs, AAPS PharmSciTech. 19 (2018) 3609–3630. https://doi.org/10.1208/s12249-018-1188-8.

  • Reviewer comment: I suggest the Authors to perform nanoemulsions next time.

Response: We thank the reviewer for this suggestion, and in fact we are working on nanoemulsions as well. That being said, our working hypothesis is, in fact, that nanoemulsions are also not going to be superior to pure vegetable oils when these oils are used in conditions that provide a proper and fair control group (with amount of water co-administered with the oil equal to the amount of water in the nanoemulsion formulation), but we will see what happens.

  • Reviewer comment: line 7, please check the author list

Response: In accordance with Reviewer’s comment, the author list has been modified in the revised manuscript (Line 7 of the revised Manuscript).

  • Reviewer comment: line 39-41, the first three references are not recent (2003, 2007); the advancements in pharmaceutical technology in the field of drug delivery systems are in continuous development, therefore more up to date reviews should be included.

Response: As we mentioned before, the older good papers are necessary to cite. But as suggested by Reviewer, a more recent review paper has been added in the revised manuscript, which is practically saying the same thing as the previous references (line 41 of the revised Manuscript):

V. Patel, R. Lalani, D. Bardoliwala, S. Ghosh, A. Misra, Lipid-Based Oral Formulation Strategies for Lipophilic Drugs, AAPS PharmSciTech. 19 (2018) 3609–3630. https://doi.org/10.1208/s12249-018-1188-8.

  • Reviewer comment: line 42, the Authors state that sesame oil is the most common oil used as excipients for lipid-based formulations. However, the reported references deal only with CBD containing formulations, but they did not mentioned CBD in the text yet. I suggest the Author to describe more in depth what lipid formulations are, which oils are used, providing examples of their employment with different drugs.

Response: Since this article is an original research paper and not an extended literature review, we do not think it is necessary to describe here a more extended review of oils and lipid-based formulations (in fact, we think doing this would skew the attention away from the aims of the work). However, as suggested by the Reviewer, we have modified the statement as ‘One of the most common lipidic core excipients used in various lipid-based formulations is different vegetable fats, for example sesame oil’ in the revised manuscript (line 42 of the revised Manuscript).

  • Reviewer comment: line 61, please provide examples of drugs whose bioavailabilities were increased by lymphatic transport, and add some references.

Response: As suggested by Reviewer, examples (halofantrine, vitamin D3 and tetrahydrocannabinol (THC)) and associated references have been provided in the revised manuscript (lines 76-77 of the revised Manuscript).

  • Reviewer comment: line 73, please described the most popular CBD formulations and the latest advancements present in the literature, highlighting the drawbacks and the positives.

Response: There is no global consensus of the ‘most popular formulation’. However, as suggested by the Reviewer, we have included a recent paper describing a clinical study of CBD in sesame oil in the revised manuscript (lines 87-88 of the revised Manuscript).

  • Reviewer comment: line 90, please add more recent works regarding the solubilizing efficiency of vitamin E TPGS (i.e. retinoic acid, resveratrol).

Response: As suggested by the Reviewer, two more recent references:

  1. G. Zuccari, S. Alfei, A. Zorzoli, D. Marimpietri, F. Turrini, S. Baldassari, L. Marchitto, G. Caviglioli, Increased Water-Solubility and Maintained Antioxidant Power of Resveratrol by Its Encapsulation in Vitamin E TPGS Micelles: A Potential Nutritional Supplement for Chronic Liver Disease, Pharmaceutics. 13 (2021) 1128. https://doi.org/10.3390/pharmaceutics13081128.
  2. G. Zuccari, S. Baldassari, S. Alfei, B. Marengo, G.E. Valenti, C. Domenicotti, G. Ailuno, C. Villa, L. Marchitto, G. Caviglioli, D-α-tocopherol-based micelles for successful encapsulation of retinoic acid, Pharmaceuticals. 14 (2021). https://doi.org/10.3390/ph14030212.
  3. have been added in the revised manuscript (line 107 of the revised Manuscript).

  • Reviewer comment: line 115, LCT is right or is MCT?

Response: Thank you for the comment, we have double checked and it is LCT.

  • Reviewer comment: Figure 2. lines 238-244. The discussion is not clear, the total amounts of tri-di-monoglycerides are different in each formulations as obviously their proportion is different in the related starting mixtures. The data should be provided considering their concentration in each formulation (recovered mg / initial mg x100).

Response: As suggested by Reviewer, we have updated the information for Figure 2, the y-axis is “Triglyceride, diglyceride or monoglyceride amount (mg)” in revised manuscript (line 333 of the revised Manuscript).

Reviewer 4 Report

Manuscript ID: pharmaceutics-1340315

The manuscript of Wanshan Feng, Chaolong Qin, Elena Cipolla, Jong Bong Lee, Atheer Zgair, Yen Ju Chu, Catherine A Ortori, Michael J. Stocks, Cris S. Constantinescu, David A. Barrett, Peter M. Fischer and Pavel Gershkovich as Co-authors: “Inclusion of medium-chain triglyceride in lipid-based formulation of cannabidiol facilitates micellar solubilization in vitro, but in vivo performance remains superior with pure sesame oil vehicle” presents the studies regarding the performance of lipid-based formulations with addition of medium-chain triglyceride or surfactants to the sesame oil vehicle in vitro and in vivo using the delivery of poorly water-soluble drug cannabidiol as a model.

Some important issues needed to be added and addressed before publication.

Major issues:

  1. This manuscript is based on the hypothesis that lipid-based formulation of cannabidiol may facilitate micellar solubilisation. This assumption may raise the question - how the authors can prove that the system is micellar? In the manuscript no physical characterisation of the obtained systems is provided. Please add data regarding characterisation of the obtained systems.

Minor issues:

  1. Lines 6-7. Is the list of author completed or not?
  2. Lines 112 and 126. The same number and title of subparagraph. Please correct.
  3. Lines 130 1nd 133. Please use subscript for 2 in the formula of CaCl2.
  4. Please use the same style points for the description of temperature values (With/without space between digits and degree).

Consequently, I do recommend accepting this manuscript for publication with major revision.

Author Response

Reviewer #4:

The manuscript of Wanshan Feng, Chaolong Qin, Elena Cipolla, Jong Bong Lee, Atheer Zgair, Yen Ju Chu, Catherine A Ortori, Michael J. Stocks, Cris S. Constantinescu, David A. Barrett, Peter M. Fischer and Pavel Gershkovich as Co-authors: “Inclusion of medium-chain triglyceride in lipid-based formulation of cannabidiol facilitates micellar solubilization in vitro, but in vivo performance remains superior with pure sesame oil vehicle” presents the studies regarding the performance of lipid-based formulations with addition of medium-chain triglyceride or surfactants to the sesame oil vehicle in vitro and in vivo using the delivery of poorly water-soluble drug cannabidiol as a model.

Some important issues needed to be added and addressed before publication.

Major issues:

  • Reviewer comment: This manuscript is based on the hypothesis that lipid-based formulation of cannabidiol may facilitate micellar solubilisation. This assumption may raise the question - how the authors can prove that the system is micellar? In the manuscript no physical characterisation of the obtained systems is provided. Please add data regarding characterisation of the obtained systems.

Response: Thank you for this comment, the work for the characterization of different phases following the in vitro lipolysis including micellar phase has been done in the past by other authors. We have introduced these references in the revised manuscript (lines 218-219 of the revised Manuscript).

  1. K.J. MacGregor, J.K. Embleton, J.E. Lacy, E.A. Perry, L.J. Solomon, H. Seager, C.W. Pouton, Influence of lipolysis on drug absorption from the gastro-intestinal tract, Adv. Drug Deliv. Rev. 25 (1997) 33–46. https://doi.org/10.1016/S0169-409X(96)00489-9.

  1. L. Sek, C.J.H. Porter, A.M. Kaukonen, W.N. Charman, Evaluation of the in-vitro digestion profiles of long and medium chain glycerides and the phase behaviour of their lipolytic products, J. Pharm. Pharmacol. 54 (2010) 29–41. https://doi.org/10.1211/0022357021771896.

  1. O. Hernell, J.E. Staggers, M.C. Carey, Physical-Chemical Behavior of Dietary and Biliary Lipids during Intestinal Digestion and Absorption. 2. Phase Analysis and Aggregation States of Luminal Lipids during Duodenal Fat Digestion in Healthy Adult Human Beings, Biochemistry. 29 (1990) 2041–2056. https://doi.org/10.1021/bi00460a012.

Minor issues:

  1. Reviewer comment: Lines 6-7. Is the list of author completed or not?

Response: As suggested by Reviewer, the author list has been modified in the revised manuscript (Line 7 of the revised Manuscript).

  1. Reviewer comment: Lines 112 and 126. The same number and title of subparagraph. Please correct.

Response: As suggested by Reviewer, the title for section 2.3 has been changed to “in vitro lipolysis” in revised manuscript.

  1. Reviewer comment: Lines 130 and 133. Please use subscript for 2 in the formula of CaCl2.

Response: As suggested by Reviewer, this change has been introduced in the revised Manuscript (lines 196 and 207 of the revised Manuscript).

  1. Reviewer comment: Please use the same style points for the description of temperature values (With/without space between digits and degree).

Response: As suggested by Reviewer, these changes have been introduced in the revised manuscript (lines 178 and 213 of the revised Manuscript).

Reviewer 5 Report

The manuscript submitted by Feng et al. deals with in vitro and in vivo assessment of several different lipid-based formulations of cannabidiol containing medium-chain triglycerides. The paper is interesting for readers in the field of pharmaceutical technology and industry and fits under the scope of the Journal. In general, it is clear and well-written. There are only a few minor suggestions that should be addressed to strengthen the manuscript.

Line 7: there is „and“ at the end of the authors’ list-please rewrite

Lines 115-120: it is not necessary to indicate description of the formulations in the text (only cite Table 1).

Line 130: instead of Na taurocholate hydrate, rather write sodium taurocholate hydrate.

Lines 130, 133: 2 in CaCl2 should be in subscript.

Line 134: should be min instead of mins, as already present in line 135.

Line 212: full stop should be at the end of the sentence.

Legend for Table 2: please consider use of smaller font size.

When discussing statistical difference in the text please consider use of the test name and p value in parentheses (lines 277-284).

Please check the font size in Figures 5 and 6 (y-axis, mg/ml, m in mg)

Lines 304 and 308: delete parenthesis after n=4.

Author Response

Reviewer #5:

The manuscript submitted by Feng et al. deals with in vitro and in vivo assessment of several different lipid-based formulations of cannabidiol containing medium-chain triglycerides. The paper is interesting for readers in the field of pharmaceutical technology and industry and fits under the scope of the Journal. In general, it is clear and well-written. There are only a few minor suggestions that should be addressed to strengthen the manuscript.

  1. Reviewer comment: Line 7: there is „and“ at the end of the authors’ list-please rewrite

Response: As suggested by Reviewer, the author list has been modified in the revised manuscript (Line 7 of the revised Manuscript).

  1. Reviewer comment: Lines 115-120: it is not necessary to indicate description of the formulations in the text (only cite Table 1).

Response: We thank the reviewer for the comment. Based on comments from other reviewers, it does look like the description of formulation in text provides some additional clarity. Therefore, we would prefer to keep it.

  1. Reviewer comment: Line 130: instead of Na taurocholate hydrate, rather write sodium taurocholate hydrate.

Response: As suggested by the Reviewer, this change has been introduced in the revised manuscript (Line 196 of the revised Manuscript).

  1. Reviewer comment: Lines 130, 133: 2 in CaCl2 should be in subscript.

Response: As suggested by the Reviewer, this change has been introduced in the revised Manuscript (lines 196 and 207 of the revised Manuscript).

  1. Reviewer comment: Line 134: should be min instead of mins, as already present in line 135.

Response: As suggested by the Reviewer, this change has been introduced in the revised Manuscript (Line 208 of the revised Manuscript).

  1. Reviewer comment: Line 212: full stop should be at the end of the sentence.

Response: As suggested by Reviewer, this change has been introduced in the revised Manuscript (Line 290 of the revised Manuscript).

  1. Reviewer comment: Legend for Table 2: please consider use of smaller font size.

Response: As suggested by the Reviewer, the font has been decreased for Table 2 legend in the revised manuscript.

  1. Reviewer comment: When discussing statistical difference in the text please consider use of the test name and p value in parentheses (lines 277-284).

Response: We thank the reviewer for the comment. We have already indicated this information in the figure legend, therefore, we would prefer not to show it in the text to avoid duplication of information.

  1. Reviewer comment: Please check the font size in Figures 5 and 6 (y-axis, mg/ml, m in mg)

Response: We thank the reviewer for the comment. We have double checked the font size and I have made the image larger for Figure 6.

  1. Reviewer comment: Lines 304 and 308: delete parenthesis after n=4.

Response: As suggested by Reviewer, the parenthesis has been deleted in the revised Manuscript (Lines 397 and 405 of the revised Manuscript).

Round 2

Reviewer 3 Report

The Authors responded to all my inquiries so no further comment is needed.

Author Response

Thank you for your kind comments

Reviewer 4 Report

Authors revised manuscript, however, lines, mentioned by authors (218-219) don't contain this information.  I repeat my question - This manuscript is based on the hypothesis that lipid-based formulation of cannabidiol may facilitate micellar solubilisation. This assumption may raise the question - how the authors can prove that the system is micellar? In the manuscript no physical characterisation of the obtained systems is provided. Please add data regarding characterisation of the obtained systems.

Author Response

Reviewer #4:

  1. Reviewer comment: Authors revised manuscript, however, lines, mentioned by authors (218-219) don't contain this information. I repeat my question - This manuscript is based on the hypothesis that lipid-based formulation of cannabidiol may facilitate micellar solubilisation. This assumption may raise the question - how the authors can prove that the system is micellar? In the manuscript no physical characterisation of the obtained systems is provided. Please add data regarding characterisation of the obtained systems.

Response: Thank you for this comment. The in vitro lipolysis system and process have been already thoroughly investigated in the past by multiple groups, including the characterization of sediment, undigested lipid and micellar phases. To clarify, in this study, we were producing CBD solution in oil mixtures rather than the micellar formulation to start with. It is only after the digestion in the gastrointestinal (GI) tract (involving action of lipases) that micelles are formed. The lipid digestion products of the oils in the intestinal lumen form mixed micelles together with bile salt and phospholipids. The in vitro lipolysis is an experimental tool to simulate the digestion process in GI tract, and the micelles that are formed distributed into the middle aqueous micellar phase by ultracentrifugal force in the model. We have provided multiple references for the characterization of micellar phase following the in vitro lipolysis that has been done in the past by other authors:   

  1. O. Rezhdo, S. Di Maio, P. Le, K.C. Littrell, R.L. Carrier, S.H. Chen, Characterization of colloidal structures during intestinal lipolysis using small-angle neutron scattering, J. Colloid Interface Sci. 499 (2017) 189–201. https://doi.org/10.1016/j.jcis.2017.03.109. In this work the characterization of micelles was performed using small-angle neutron scattering (SANS) technique coupled with in vitro lipolysis model, with the scattering vector Q (Å-1) being a qualitative parameter for the micelles. This study has shown that when Q is in a region of 0.01-0.5 Å-1 , it indicates the presence of mixed micelles during the lipid digestion in in vitro lipolysis system, and the micelles size changes from Ra 2.9 nm to 9.5 nm during 120 min. They found that triglycerides, the mixture of pancreatin, inhibitor, and methanol have no effect on the structure and number of bile micelles, but the digested lipids (monoglycerides and fatty acids) are integrated with bile micelles and lead to the increase in the size of the micelles during the in vitro lipolysis.
  2. J.Ø. Christensen, K. Schultz, B. Mollgaard, H.G. Kristensen, A. Mullertz, Solubilisation of poorly water-soluble drugs during in vitro lipolysis of medium- and long-chain triacylglycerols, Eur. J. Pharm. Sci. 23 (2004) 287–296. https://doi.org/10.1016/j.ejps.2004.08.003. The authors in this work assessed the aqueous layer samples following in vitro lipolysis by dynamic light scattering (DLS). Figure 2 in this work shows that the micellar size (hydrodynamic radius, RH) changes from 3 to 7.8 nm in aqueous phase following LCT lipolysis.
  3. A.T. Larsen, P. Sassene, A. Müllertz, In vitro lipolysis models as a tool for the characterization of oral lipid and surfactant based drug delivery systems, Int. J. Pharm. 417 (2011) 245–255. https://doi.org/10.1016/j.ijpharm.2011.03.002.’ This review paper described the characterization of in vitro lipolysis samples in section 3 of that work. Figure 2 in that article has shown that the structure changes during the digestion of lipid-based formulation in in vitro lipolysis system using Cryogenic transmission electron microscopy (Cryo-TEM). This is in accordance with a previous study by D.G. Fatouros, B. Bergenstahl, A. Mullertz, Morphological observations on a lipid-based drug delivery system during in vitro digestion, Eur. J. Pharm. Sci. 31 (2007) 85–94. https://doi.org/10.1016/j.ejps.2007.02.009. At time point zero, intact oil droplets and micelles with diameter approximately 10 nm originating from the lipolysis medium were present, then the unilamellar and bi-lamellar vesicles (diameter from 20 nm up to 200 nm) appeared after 2 min. There were less intact oil droplets and unilamellar vesicles observed in 30 min, and more micelles at around 10 nm diameter were present in the in vitro lipolysis samples. The zeta-potential values were reduced from -8 mV to around -11 mV during the digestion of formulation in in vitro lipolysis system at 25 â—¦C, whereas the zeta-potential values and particle size for bile micelles without lipid formulation presetn were −20.3 mV and 5 nm, respectively.

We provide here the overview of this literature to answer Reviewer’s questions, but we do not think this information should be introduced into the Manuscript, as this is work that was done previously by other authors and has little relevance for the current work. We have introduced these references in the revised manuscript (lines 150-152 of the revised Manuscript). This sentence is the last paragraph of section 2.3 states: The final lipolysis medium was separated into three layers (sediment, micellar and undigested lipid layer) by means of ultra-centrifugation at 268,350 g, 37 °C for 90 min [36–38].